# The Effect of Different Storage Conditions on Phytochemical Composition, Shelf-Life, and Bioactive Compounds of Voghiera Garlic PDO

**DOI:** 10.3390/antiox12020499

**Published:** 2023-02-16

**Authors:** Paola Tedeschi, Federica Brugnoli, Stefania Merighi, Silvia Grassilli, Manuela Nigro, Martina Catani, Stefania Gessi, Valeria Bertagnolo, Alessia Travagli, Maria Fiorenza Caboni, Alberto Cavazzini

**Affiliations:** 1Department of Chemical, Pharmaceutical and Agricultural Sciences, University of Ferrara, 44121 Ferrara, Italy; 2Department of Translational Medicine, University of Ferrara, 44121 Ferrara, Italy; 3Department of Agricultural and Food Sciences and Technologies, University of Bologna, 40127 Bologna, Italy

**Keywords:** Voghiera garlic PDO, storage conditions, sulfur compounds, polyphenols, antioxidant activity, shelf-life, breast cancer cells, anti-inflammatory activity

## Abstract

Voghiera garlic is an Italian white garlic variety which obtained in 2010 the Protected Designation of Origin. It is widely used for culinary purposes or as an ingredient for supplement production due to its phytochemical compositions. The storage conditions seem to be crucial to retain the high quality of garlic bulbs and their by-products, taking into account the high importance of organosulfur and phenolic compounds for the bioactive potency of garlic and its shelf-life. This study aims to examine the effect of storage on the phytochemical composition, biological effects, and shelf-life of Voghiera garlic PDO. In detail, we considered (i) −4 °C (industrial storage) for 3, 6, and 9 months; (ii) +4 °C for 3 months (home conservation), and (iii) −4 °C for 3 months, plus +4 °C for another 3 months. We focused our attention on the organosulfur compounds, total condensed tannins, flavonoids, phenolic compounds, and related antioxidant activity changes during the storage period. To evaluate the bioactive effects, the Voghiera garlic extracts at different storage conditions were administered to a breast cancer cell line, while antioxidant and anti-inflammatory activity was detected using macrophage RAW 264.7 cells. We observed a decrease in sulfur compounds after 6 months which correlated to a decrease in bioactive effects, while the number of antioxidant compounds was stable during the storage period, showing the good effect of refrigerated temperature in maintaining garlic bulb shelf-life.

## 1. Introduction

Garlic (*Allium sativum* L.) has been widely used as food as well as dressing for foods (as a spice and for flavoring foods) since ancient times [1,2].

Numerous studies and laboratory data have validated many of the biological and medical properties attributed to this vegetable, it is helpful to decrease the risk of cardiovascular disease because of its hypocholesterolemic, anti-hypertensive, and hypolipidemic effects [3,4].

More recently, garlic has been used as a functional food to inhibit the growth of pathogens but also served in killing cancer cells in several tumors acting at a different stage of carcinogenesis, and high garlic consumption seem to be protective against various solid tumors, including breast cancer [5,6,7,8,9,10]. Moreover, the healthy properties of garlic have been characterized in terms of neuroprotection for Alzheimer disease’s therapy [4,11].

All these activities are mainly attributed to the presence of phytochemicals, which make garlic a healthy food. These compounds are first of all organosulfur compounds (OSCs), synthetized by the plant using sulfate absorbed by the roots. Sulfur compounds consist of more than 20 different biosynthetic derivatives, generated when garlic tissue is broken, starting from the thiosulfate (the most abundant is allicin), which is not stable and very reactive, which became polysulphides, the last set of transformation compounds [12].

Among these more stable OSCs, vinyldithiins (VD), diallyl sulfide (DAS), diallyl disulfide (DADS), and diallyl trisulfide (DATS) are known for their important biological activities [2,9,13,14,15].

The biological activities of raw garlic seem to be connected also to another important phytochemical, the polyphenols and antioxidant compounds [16,17]. The antioxidants of garlic are important for human health, for the ability to radical scavenging, and also for food preservation. In fact, in the food industry, garlic is used as a food additive for controlling rancidity development, reducing the formation of toxic oxidation products, and extending the shelf-life of products [2,18,19,20].

In Italy, only two garlic ecotypes have received the Protected Designation of Origin (PDO): white garlic “Polesano” and Voghiera garlic, which were awarded PDO designation in 2010 (GUUE L 126, 22-05-2010) [21].

Voghiera is a little town (Emilia Romagna, Italy), where more than 100 hectares are dedicated each year to the cultivation of Voghiera garlic PDO, with a production, on average, of 10.000 kg of dried garlic per hectare farmed.

After harvesting, bulbs and stems are dried and packaged by hand according to tradition (guaranteed by the Consortium of Voghiera Garlic Producers), sold and used fresh from August–September, or after different storage periods in a well-ventilated space at −3/−4 °C.

The Allium ecotypes, the agricultural techniques, and the operative conditions during processing and storage strongly influence the chemical composition [22]; the bioactive compounds and related beneficial effects are strictly connected to organosulfur compounds and antioxidants fraction, and taking into account that garlic plants are considered as a condiment but also for medical purposes, monitoring the storage conditions is proved to be of major importance [23].

In the present work, an extensive investigation was conducted on Voghiera garlic PDO samples stored at different temperatures for short, medium, and long periods to evaluate the quali-quantitative changes of the chemical metabolites, compared with fresh garlic. The sulfur compounds were identified and quantified from Voghiera garlic samples by GC/MS, whereas the flavonoids, the total condensed tannins, and polyphenols were detected using colorimetric reactions, and the intrinsic antioxidant activity was measured through a DPPH assay. 

On the basis of our previous results demonstrating that garlic extracts prevent the malignant progression of breast tumor cells [7], we aimed to establish the effects of Voghiera garlic extracts (fresh garlic and garlic at different storage conditions), tested for the chemical composition, on an MCF7 breast tumor cell line. We evaluated the inhibition of cell cycle progression and the expression of E-cadherin, a marker of the epithelial–mesenchymal transition (EMT), a crucial phenomenon underlying tumor progression. 

Subsequently, as the garlic extract due to its constituents presents anti-tumoral, antioxidant, and anti-inflammatory effects [3,4], experiments have been designed to address these aspects, evaluating these properties at different storage conditions. The antioxidant activity was investigated by the H_2_DCFDA test in LPS-stimulated RAW 264.7 macrophages, while the anti-inflammatory effect was tested in RAW 264.7 cells with the Nitrate/Nitrite Colorimetric Assay. 

To our knowledge, this is the first time that a multi-methodological approach has been applied to phytonutrients and healthy properties of Voghiera garlic PDO, evaluating not only fresh products but also stored garlic at different time/temperature combined conditions. Since garlic is commercially available as a raw vegetable for culinary purposes but also as a powder, a dietary supplement, and an extract for medical uses, the purpose of the work was to focus attention on possible phytochemical changes during storage periods and its correlation with biofunctional effects.

## 2. Materials and Methods

### 2.1. Plant Material

Plant material consisted of bulbs of Voghiera garlic PDO, cultivated in the PDO area with production rules, according to “Disciplinare di Produzione” and guaranteed by the Consortium of Voghiera Garlic Producers. 

Samples were collected fresh, after a natural drying process and after different storage periods, in a well-ventilated space at the Consortium of Voghiera Garlic Producers.

We analyzed garlic samples that were:Fresh (only naturally dried).Stored at a refrigerated temperature (+4 °C) for 3 months (3 M +4 °C).Stored in a well-ventilated cold room (−3/−4 °C) for 3 months (3 M −4 °C).Stored in a well-ventilated cold room (−3/−4 °C) for 6 months (6 M −4 °C).Stored in a well-ventilated cold room (−3/−4 °C) for 3 months and another 3 months at a refrigerated temperature (+4 °C) (3 M −4 °C + 3 M +4 °C).Stored in a well-ventilated cold room (−3/−4 °C) for 9 months (9 M −4 °C).

After each period, garlic samples were provided by the Consortium of Voghiera Garlic to our laboratory for analysis. 

Ethanol, diethyl ether, dimethyl sulfoxide (DMSO), and other solvents used were obtained from Carlo Erba Reagents (Milan, Italy). Chemical standards were purchased from Merck (Milan, Italy). All the solvents and chemical standards were analytical-grade products used without further purification.

### 2.2. Hydro-Alcoholic Extraction

Voghiera garlic PDO extract (VGE) was prepared in a hydro-alcoholic solution, according to Petrovic and Brugnoli [7,24]. Briefly, 350 g of Voghiera garlic fresh or at different storage conditions, provided by “Aglio del Nonno S.r.l, Consorzio produttori aglio di Voghiera DOP, Ferrara, Italy”, was crushed in 250 mL 40% ethanol for 5 days in dark conditions at a refrigerated temperature, shaking constantly. After the extraction period was subjected to centrifugation at 2500× *g* for 10 min at 4 °C, Supernatant (VGE) was stored at −20 °C until use.

### 2.3. Diethyl Ether Extraction

The diethyl ether Voghiera garlic PDO extract (D-VGE) was prepared by adding 200 mL of diethyl ether to 100 g of crushed Voghiera garlic (fresh or at different storage conditions, provided from “Aglio del Nonno S.r.l”, Consorzio produttori aglio di Voghiera DOP, Ferrara, Italy) for 48 h in dark conditions and then subjected to centrifugation at 2500× *g* for 10 min at 4 °C. Supernatant (1 µL) was injected in GC-MS for OSC analysis and the remaining was dried and recovered with 3 mL of dimethyl sulfoxide. 

### 2.4. Organosulfur Compounds Identification and Quantification

A total of 1 µL of VGE (treated before with diethyl ether 1:1 *v*/*v*, for 24 h) was injected in the GC/MS system (Varian 3900 gas chromatograph coupled to Varian Saturn 2100 MS/MS ion trap mass spectrometer, Varian, Palo Alto, CA, USA) for the organosulfur compounds analysis. 

The OSC separation was carried out in a 60 m × 0.25 mm i.d., 25 µm film thickness, Zebron ZB-Wax capillary column (Phenomenex) with a column pressure of 10 psi and supplied with a helium carrier (flow rate of 1 mL/min.). The oven temperature program was the following: start at 50 °C for 1 min, ramp to 150 °C at 1 °C/min, and hold for 1 min, while the injector temperature was 200 °C. The MS temperature conditions were trap, 180 °C; transfer line, 200 °C; and manifold, 70 °C, and the MS acquisition data. The MS acquisitions were performed by electron ionization (EI) in full scan mode, a scan time of 1 s/scan, and an emission current of 10 µA. 

The mass spectrometer operated in scan mode (40–650 *m*/*z*) and the NIST MS library was used to evaluate the data and organosulfur compounds identification [7].

Peak quantification was obtained using reference standard diallyl disulfide (>95% GC), R^2^ = 0.9982.

Organosulfur compounds in D-VGE were detected, identified, and quantified by injecting directly the extract in diethyl ether in the GC/MS system as described above. 

### 2.5. Flavonoid Content

Total flavonoid content was determined using a colorimetric method. A total of 100 µL of VGE was added to 150 µL of NaNO_2_ solution and allowed to stand for 5 min, and then 300 µL of 10% AlCl_3_ solution and 1 mL of NaOH 1 M were added, and the final volume was adjusted to 5 mL. The absorption was detected at 510 nm vs. the reagent blank using a Beckman DU730 UV–Vis spectrophotometer (Beckman Coulter, Pasadena, CA, USA).

The calibration curve was 1–10 ppm (R^2^ = 0.9983) and the total amount of flavonoids was expressed as (+)-catechin equivalents and (µg of (+)-catechin equivalents/g of garlic). 

### 2.6. Total Condensed Tannins

Total condensed tannins were detected using a colorimetric method [25]. Briefly, 200 µL of VGE was added to 3 mL of vanillin (4% in MeOH, *w*/*v*) and 1.50 mL of HCl, and the final volume was adjusted to 5 mL with methanol. 

The absorption was measured at 500 nm vs. the reagent blank, using a Beckman DU730 UV–Vis spectrophotometer. The calibration curve was 0.5–10 ppm (R^2^ = 0.9988) and the amount of total condensed tannins is expressed as (+)-catechin equivalents (µg of (+)-catechin equivalents/g of garlic).

### 2.7. Total Phenolic Content

Total phenolic content was determined using the Folin–Ciocalteu method as described by Singleton [26], partially modified. A total of 500 µL of Folin–Ciocalteu reagent was added to 50 µL of VGE and stood for 5 min in the dark. Then, 2 mL of a 10% aqueous Na_2_CO_3_ solution was added and the final volume was adjusted to 10 mL. The solution was allowed to stand for 90 min at room temperature and in the dark. The measurement was at 700 nm vs. reagent blank, using a Beckman DU730 UV–Vis spectrophotometer. The calibration curve was 0.5–10 ppm (R^2^ = 0.9974) and the total phenolics are expressed as gallic acid equivalents (µg of gallic acid equivalents/g of garlic). 

### 2.8. DPPH Scavenging Assay

The antioxidant capacity was tested with a 2,2-diphenyl-1-picrylhydrazyl (DPPH) assay, according to the method of Fukumoto [27] with minor modifications. Trolox (6-hydroxy-2,5,7,8-tetramethylchroman-2-carboxylic acid), 0.05–1 mM in methanol, was used to prepare a calibration curve. A deep purple solution of DPPH 2,2-diphenyl-1-picrylhydrazyl (0.06 mM) was prepared in methanol and the absorbance was measured at 515 nm, using a Beckman DU730 UV–Vis spectrophotometer. 

Aliquots of 50 µL of VGE garlic were added to 1450 µL of the DPPH methanol solution; the mixture was stirred vigorously and kept for 15 min in a dark at room temperature. The decrement of spectrophotometric absorbance with the color decrease toward yellow was registered. The antioxidant activity was calculated by the percentage of inhibition of the DPPH radical:Inhibition% = [Control Abs (t = 0 min) − Sample Abs (t = 15 min)]/Control Abs (t = 0 min) × 100
where Control Abs was the absorbance of the control (DPPH) and sample Abs was the absorbance of the standard or the sample. 

The calibration curve (R^2^ = 0.9914) was expressed as mM of Trolox equivalents.

### 2.9. Cells and Treatments

The MCF7 breast cancer-derived cell line was from the American Type Culture Collection (Rockville, MD, USA). Cells were grown in Dulbecco’s modified Eagle’s medium (DMEM, Gibco Laboratories, Grand Island, NY, USA) supplemented with 10% fetal bovine serum (FBS, Gibco Laboratories) and 1% penicillin-streptomycin (Gibco Laboratories) at 37 °C in a humidified atmosphere of 5% CO_2_ in the air, and monthly tested for mycoplasma and other contaminations.

A 1:800 dilution of VGE derived from garlic after different storage conditions was administered to MCF7 cells for 72 h. As extracts contained ~22% ethanol [24], a corresponding ethanol dilution (vehicle) was used in untreated conditions.

Cells in all experimental conditions were subjected to evaluation of viability by using the Trypan Blue Exclusion Test, in which cells were suspended in PBS containing trypan blue and then examined with an inverted phase-contrast microscope (Diaphot, Nikon, Melville, NY, USA).

RAW 264.7 macrophage murine cells (BS TCL 177, IZSLER Biobank, Brescia, Italy) were grown in DMEM containing high glucose and 10% heat-inactivated fetal bovine serum (FBS), 1% of a penicillin (100 U/mL) and streptomycin (100 g/mL) mixture. To maintain the ideal confluence (80%), cells were divided three times a week in a humidified atmosphere with 5% CO_2_ and 37 °C temperature. To induce the proinflammatory response, RAW 264.7 were treated for 24 h with 1 µg/mL of lipopolysaccharide (LPS) (derived from Escherichia coli, serotype 055:B5, soluble in cell culture medium). Other treatments included adding various VGE dilutions 30 min before LPS. The cell medium was replaced with a serum-free medium prior to each experiment.

### 2.10. Cell Cycle Analysis

MCF7 cells under different experimental conditions were analyzed for cell cycle distribution by using a FACS Calibur flow cytometer (BD Biosciences, San Josè, CA, USA), following a previously described procedure [7]. The percentage of cells in the G0/G1, S, and G2/M phases was calculated by the CellQuest Pro 6.0 software (BD Biosciences), as previously reported [7].

### 2.11. Immunochemical Analysis

Total cell lysates (15 µg of proteins) from MCF7 cells cultured in the different experimental conditions were separated on 7.5% polyacrylamide denaturing gels and blotted to nitrocellulose membranes (GE Healthcare Life Science, Little Chalfont, UK), that reacted with antibodies directed against E-cadherin (Santa Cruz Biotechnology, Santa Cruz, CA, USA) and β-Tubulin (Sigma Aldrich, Milan, Italy), following previously reported procedures [28]. The immunocomplexes were detected by using the ECL system (Perkin-Elmer, Boston, MA, USA) and the chemiluminescence signals were acquired with an ImageQuantTM LAS 4000 imager (GE Healthcare Life Science, Little Chalfont, UK), and quantified by means of Image Quant TL software (GE Healthcare Life Science, Little Chalfont, UK), as previously reported [7].

### 2.12. MTS Assay

According to the manufacturer’s instructions from the CellTiter 96 AQueous One Solution cell proliferation assay, the MTS test was carried out to assess the health of RAW 264.7 macrophages (Promega, Milan, Italy). In total, 30,000 cells were seeded into 96 multiwell plates, which were subsequently filled with 100 μL of complete media in the absence and presence of various VGE dilutions for 24 h. The cells were then allowed to attach overnight. The MTS solution was applied to each well at the conclusion of the incubation time. At 570 nm, the optical density of each well was measured using a spectrophotometer.

### 2.13. H_2_DCFDA Assay

By using the 2′,7′-dichlorofluorescein diacetate (H2DCFDA) assay in RAW 264.7 macrophage cells, the antioxidant capacity of a 1:1000 VGE dilution was examined [29]. Specifically, 30,000 cells were seeded in a 96-well black plate and cultured there for a whole night. Treatments were then carried out in a medium free of serum. Each well’s supernatant was withdrawn after 24 h, and 100 µL of a 10 µM H2DCFDA solution was then added. The plate was then incubated at 37 °C in the dark. Three PBS washes were carried out after one hour, and then 100 µL of PBS was given to each well. With the Ensight multimodal plate reader, the fluorescence was read at 485 nm for excitation and 538 nm for emission (Perkin Elmer, Milan, Italy).

### 2.14. Nitric Oxide Assay

Vinci Biochem’s Nitrate/Nitrite Colorimetric Assay Kit was used to examine the 1:1000 VGE dilution’s ability to reduce inflammation in RAW 264.7 cells (Florence, Italy) [29]. Specifically, 80 µL of the supernatants from each well in a 24-well plate containing 150,000 cells were transferred to a 96-well plate together with 10 µL of nitrate reductase and 10 µL of its cofactor. The two Griess reagents were added after 2 h of incubation, transforming the whole nitrite into a purple azoic molecule. The Ensight multimodal plate reader (Perkin Elmer, Milan, Italy) was used to measure absorbance with the wavelength set to 550 nm. Nitrate was used in the standard curve, which made it possible to calculate the concentration of nitrate + nitrite, which is proportional to the red absorption.

### 2.15. Statistical Analysis

The results of three independent experiments were expressed as means ± standard deviation or means ± standard errors. When necessary, one-way analysis of variance (ANOVA) and Dunnett’s test were used to assess data sets. A *p*-value of 0.05 or less was regarded as statistically significant.

## 3. Results and Discussion

The phytochemistry of garlic has been the subject of thousands of research papers in the last decade and several extraction techniques are described in the literature using different solvents, water, ethanol, methanol, acetone, and hexane diethyl ethers, based on the different molecules to be extracted. 

The extraction protocol applied in the present study was based on the use of a hydro-alcoholic mixture (water/ethanol, in a ratio of 60/40), at a refrigerated temperature, already reported by Petrovic et al. and Brugnoli et al. [7,24] as an efficient extraction method.

The hydro-alcoholic mixture allowed us to obtain an extract rich both in organosulfur compounds and antioxidant fractions, usable in cell tests, and a “sustainable” method with “green” solvents.

As already described by the literature [1,2,15], garlic bulbs are rich in phytochemicals, which make garlic a healthy food; but sensory and biological characteristics connected to phytochemical molecules can be conditioned by storage treatments in addition to the variety, agricultural conditions, and specific extraction [30].

### 3.1. Organosulfur Compounds

When the garlic tissue is broken, the first organosulfur compounds generated are the thiosulfinates, and allicin is the most abundant, which confers the pungency of garlic bulbs [2,3,31]. On the other hand, allicin is an unstable molecule and highly reactive, and for this reason, it transforms into a series of OSCs that are more stable, including the polysulfides, the last set of transformations.

To detect the OSC garlic profile and evaluate the storage effects, the hydro-alcoholic extracts obtained from fresh garlic and garlic after different storage conditions were submitted to GC/MS analysis.

The GC/MS analysis identified 14 different OSCs with a retention time between 13 to 87 min (Figure 1).

Among the OSCs revealed, vinyldithiins (3-vinyl-1,2-dithiacyclohex-4 ene and 3-vinyl-1,2-dithiacyclohex-5 ene, peak n. 12 and 14), and Diallyl Disulfide (peak n. 6) were the most represented and have been mentioned for their important biological activity. 

Taking into account the several uses of garlic bulbs, ensuring proper and safe storage conditions is proved to be of major importance. As shown by the literature [3,23,32], storage conditions are one of the factors that influence OSC content in garlic, closely related to the type of the cultivar and pre-harvest conditions.

Proper storage conditions are important to maintain the high quality of garlic cloves and related by-products, especially considering the high instability of OSCs.

The changes of organosulfur compounds in Voghiera garlic PDO cloves during storage at different temperatures for 3, 6, and 9 months are shown in Table 1.

The data confirmed the large amounts of vinyldithiins and diallyl disulfide, and other OSCs have also been found but in a lower concentration.

Examining different storage periods, the levels of two vinyldithiins, diallyl disulfide, and other OSCs of different storage garlic extract used in this study showed that the OSCs have been maintained during the first three months of storage at both temperature conditions.

Only some minor OSCs (diallyl sulfide, di-tert dodecyl disulfide, and 1-propenyl allyl disulfide) showed a marked decrease already after the first 3 months of storage. However, it is interesting to observe a statistically significant decrease, which was particularly important after six months of storage for each OSCs compounds analyzed. Focusing the attention on the total content of sulfur compounds, it was even six times lower in samples stored for more than six months, highlighting that the bioactive organosulfur compounds of garlic “suffer” long time storage even if at a refrigerated temperature.

### 3.2. Antioxidant Compounds

The important biological activities of garlic are mainly attributed to the high content of phytochemical compounds, sulfur compounds already described above, and polyphenols.

Antioxidants in garlic are considered and studied for their capacity to protect the human body against reactive oxygen species and relative oxidative damage [33]; but they are also important in the food industry because garlic can be used as an additive for controlling oxidation development, extending the shelf-life of products and maintaining their nutritional quality [2,19,34].

Due to the consumer’s growing interest in the use of natural products, the use of plant sources as food natural preservatives in the food industry is increasing [35,36].

In this perspective, the use of garlic may increase the shelf-life of foods because of its antibacterial and antioxidant action, and it is applied in the food industry, especially in processed meat products [35,37]. Raeisi et al. [38] reported good results of fortification of chicken nuggets with encapsulated fish oil to improve omega-3 polyunsaturated fatty acids combined with garlic essential oil to delay chemical deterioration, decelerate microbial growth, and extend the shelf-life of the samples during refrigerated storage.

Moreover, the presence of polyphenols and antioxidant compounds in garlic is also important to maintain the intrinsic characteristics and to ensure a better shelf-life of the product itself.

The Voghiera garlic hydro-alcoholic extract was analyzed to quantify the antioxidant compounds evaluating possible storage-related changes.

Figure 2 showed the flavonoids, tannins, polyphenols, and the related antioxidant activity in extracts from Voghiera garlic stored under different conditions. 

Flavonoids and tannins are expressed as +(−) catechin Equation, while total phenols as Gallic acid eq. The antioxidant activity was measured by the radical scavenging of DPPH radical and was quantified as µM/Trolox Eq.

The analysis of variance evidenced the higher concentration of flavonoids in VGE obtained from fresh garlic samples; but even if significant for * *p* < 0.05 vs. fresh garlic, the flavonoid content in storage samples was preserved during the storage period until 9 months, at both refrigerated temperature conditions.

Tannins instead showed a fluctuating trend, with higher concentrations after storage at −4 °C for 3 months and after six months maintained first at cold temperature and after three months at a simply refrigerated temperature (+4 °C). Considering only the storage at cold temperatures (−4 °C), the data presented a decrease over time. It is interesting to focus attention on conservation at +4 °C for 3 months. The results seem to indicate a loss of tannins in three months at refrigerated temperatures, while the same time storage at a cold temperature maintained these compounds.

Finally, VGE samples exhibited a good quantity of total phenolic compounds and good antioxidant activity (Figure 2), and contrary to sulfur compounds, the results obtained showed that polyphenol and antioxidant activity of Voghiera garlic PDO remained stable up to 9 months of storage at −4 °C, but also after 6 months, of which only 3 were at a refrigerated temperature.

Moreover, polyphenols present in VGE samples showed a positive and important correlation (Pearson correlation coefficient 0.60) with free radical scavenging activity.

These results have been also corroborated by the literature [39,40,41], which stated that organosulfur compounds decreased over the storage period time, while polyphenols and antioxidant activity enhanced.

Furthermore, the same authors reported that the antioxidant activity of garlic was at a maximum after 8 weeks of storage at 20 °C. In our study, Voghiera garlic PDO was stored always at a refrigerated and/or cold temperature. Flavonoids, polyphenols, and antioxidant activity seem unaffected by the two different temperatures and the contents have been maintained over time. On the contrary, tannins appeared more susceptible to temperature and time conditions with a decrease after 3 months at +4 °C, and after 6 and 9 months at cold temperatures. 

### 3.3. Bioactive Effects—The Anti-Tumoral Effect of VGE on Breast Cancer Cells 

Although the molecular mechanism at the basis of garlic effects is not completely understood, its oil-soluble OSCs, such as diallyl sulfide (DAS), seems to represent the major responsible for the anticancer properties of garlic [42]. Garlic derivatives may affect different stages of carcinogenesis, including the activation of oxidative enzymes, the proliferation of clonal cells, and the growth and invasion of tumor cells [42,43]. In this context, we have previously demonstrated that VGE shows anti-cancer properties in both invasive and non-invasive breast tumor cells, promoting an epithelial phenotype and reducing cell growth, and may exert a protective role against the transition from epithelial to mesenchymal phenotype induced by low oxygen availability, highlighting the potential involvement of Voghiera hydro-alcoholic garlic extract in new approaches for preventing progression of breast tumors [7]. Here, we evaluated the effects of garlic conservation on the antitumor role of VGE, showing that accumulation of MCF7 cells in S and/or G2/M phases is observed in all conditions but tends to decrease with extracts from garlic after storage of at least 6 months (Figure 3A). 

Additionally, the increase in E-cadherin, indicating the role of VGE in promoting an epithelia phenotype, tends to decrease in samples treated with extracts from garlic stored for at least 6 months (Figure 3B,C), according to the total organosulfur amount. Furthermore, both the accumulation in G2/M and the increase in E-cadherin do not occur or are modest in the samples treated with extracts from garlic stored for 3 months at +4 °C, possibly correlated with the appearance of derivative products, allowing us to assess that the methods of storage of garlic influence the effectiveness of its extract and indicating that a temperature of −4 °C is more suitable for storage. 

### 3.4. Antioxidant Properties of VGE on RAW 264.7 Macrophage Cells

The antioxidant and anti-inflammatory potential of VGE was tested in RAW 264.7 macrophage cells by a DCFDA and nitric oxide (NO) assay. RAW 264.7 cells are macrophage-like, Abelson leukemia virus-transformed cell lines obtained from BALB/c mice. This study has made use of this cell line as a typical and established model of mouse macrophages for the investigation of cellular reactions to microorganisms and their metabolites. Briefly, DCFDA is used for the determination of reactive species at the cellular level both in the intracellular space. DCFDA is not a fluorescent molecule but in the cellular environment, it is deacetylated into dichlorofluorescin (DCFH) by esterase enzymes. It is reactive toward reactive species forming DCF, a strongly fluorescent molecule. The NO test allows us to accurately determine, through the Griess reaction, the concentration of nitrites and nitrates (NO) released by the cell following a pro-inflammatory insult. The results of the study showed that RAW 264.7 cells treated with LPS 1 µg/mL for 24 h increased the release of free radicals and NO. The hydro-alcoholic fresh VGE tested at a 1:1000 dilution did not show statistically significant effects (Figure 4A,B), and smaller dilutions (1:10, 1:100, 1:500) were not tested due to the toxicity of the VGEs revealed by the MTS Cell Viability Test performed on RAW 264.7 cells for 24 h.

To investigate these unsatisfactory results even with fresh garlic, rich in OSCs and antioxidant molecules, according to the same literature data which use different solvents for the extraction of allicin and related derivatives [14,44], we decided to extract the sulfur compounds fraction with diethyl ether. 

We extracted sulfur compounds from Voghiera garlic PDO at all storage times with diethyl ethers and characterized the different extracts in a sulfur compound composition and quantity as already described for hydro-alcoholic extracts. The OSCs profile (Table 2) was the same as hydro-alcoholic extractions, again with a large amount of the two vinyldithiins and with an important decrease in sulfur compounds during storage periods, already evident after 3 months of storage but with a higher amount of OSCs in absolute value than hydro-alcoholic extracts. 

The diethyl ether extracts after dryness were reconstituted in DMSO and tested to evaluate the antioxidant and anti-inflammatory effects. 

The fresh D-VGE obtained in DMSO at a 1:1000 dilution revealed a significant reduction in the levels of free radicals and NO induced by LPS (Figure 4A,B). 

Interestingly, no significant differences were found with regards to the effects of D-VGEs compared to those deriving from garlic stored for 3, 6, and 9 months at −4 °C, for 3 months at a temperature of +4 °C; stored at −4 °C for 3 months and subsequently kept for 3 months at a temperature of +4 °C (Figure 5A,B). The data seemed to indicate the important role of sulfur compounds in reducing the levels of free radicals and the anti-inflammatory effect induced by LPS, even in long time storage time, if present in large quantities. 

## 4. Conclusions

Voghiera garlic PDO samples proved to have significant and interesting antioxidant activity over long storage periods at a refrigerated temperature. These results seemed to be strictly connected to flavonoids and polyphenols in VGE which were maintained or even increased over time, guaranteeing a good shelf-life for the garlic bulbs and cloves. 

Organosulfur compounds, contained in large quantities in fresh garlic, “suffer” long storage times, with an important loss after 6 months. These data greatly affect the important biological properties, such as preventing breast cancer progression and reducing the pro-inflammatory effect induced by LPS, which seem to be strictly connected to the high amount of OSCs contained in the VGE.

Refrigerated temperature (−4 °C) used by Voghiera garlic PDO companies preserved this important product for the entire sales period, but quantifying the OSCs and antioxidant compounds over time is of fundamental importance for the possible employment of this interesting vegetable in nutraceutical, pharmacological, and food supplement sectors where a standardized phytochemicals concentration is absolutely required. Furthermore, this knowledge may be useful for garlic producers to evaluate the best method of preserving this vegetable, guaranteeing as much as possible all the important chemical compounds.

## Figures and Tables

**Figure 1 antioxidants-12-00499-f001:**
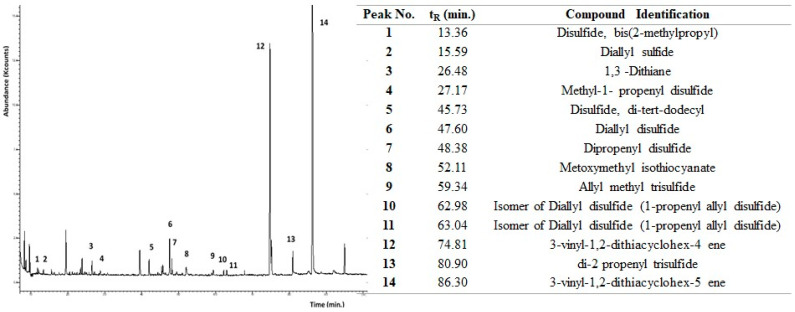
GC/MS chromatogram of OSCs detected in VGE fresh extract and the identification of the major peaks using EI mass spectra information and NIST library data.

**Figure 2 antioxidants-12-00499-f002:**
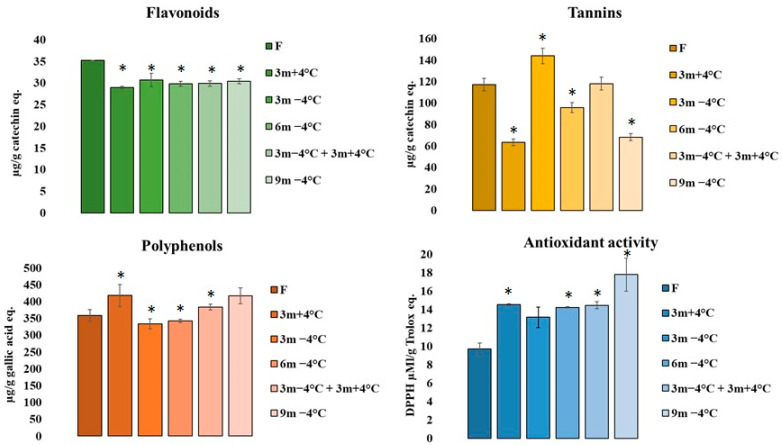
The total content of flavonoids, tannins, polyphenols, and antioxidant activity in Voghiera garlic PDO under different storage conditions. The results were expressed as means (n = 3) ± SD. * *p* < 0.05 vs. fresh garlic.

**Figure 3 antioxidants-12-00499-f003:**
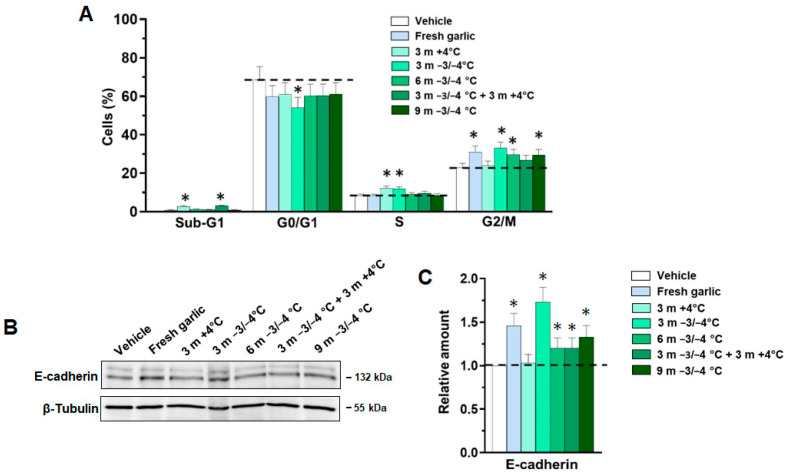
Anti-tumoral effects of VGE from garlic under different storage conditions on breast cancer-derived cells. In (**A**), the cell cycle distribution of MCF7 cells grown for 72 h in control conditions (Vehicle) or the presence of 1:800 VGE from garlic conserved at the indicated storage conditions. * *p* < 0.05 with respect to Vehicle. In (**B**), representative immunoblot analysis was performed with the indicated antibodies of lysates from MCF7 cells cultured, as indicated above. In (**C**), histograms, as deduced from the analysis of chemiluminescence signals, reported the levels of E-cadherin normalized to β-Tubulin, used as an internal control for equivalence of loaded proteins. * *p* < 0.05 with respect to control conditions taken as 1. Dotted lines indicate the values of the control conditions (vehicle). All data are the mean of three separate experiments ± SD.

**Figure 4 antioxidants-12-00499-f004:**
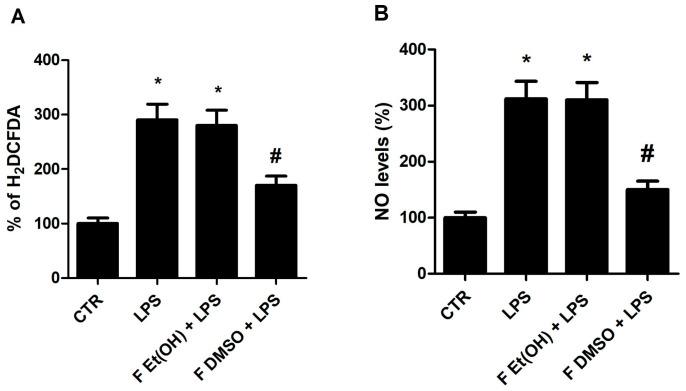
The effect of VGEs and D-VGEs on the modulation of LPS-stimulated oxygen radical release (DCFDA) (**A**) and NO (**B**) in macrophage RAW 264.7 cells. * *p* < 0.01 versus control (CTR). # *p* < 0.01 toward LPS. Analysis by ANOVA followed by Dunnett’s test, N = 3.

**Figure 5 antioxidants-12-00499-f005:**
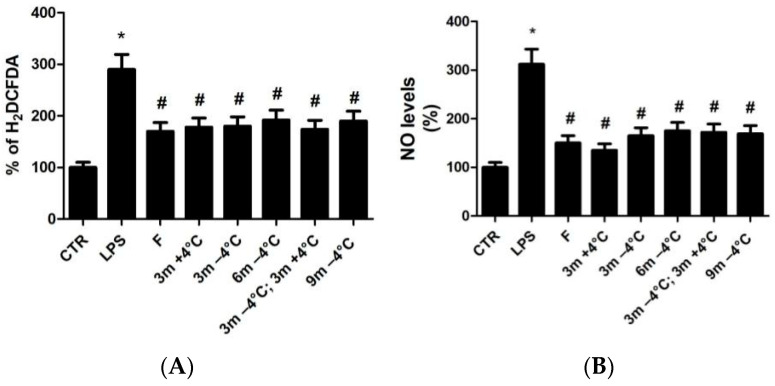
The effect of D-VGEs at different storage conditions on the modulation of LPS-stimulated oxygen radical release (DCFDA) (**A**) and NO (**B**) in macrophage RAW 264.7 cells. * *p* < 0.01 versus control (CTR). # *p* < 0.01 toward LPS. Analysis by ANOVA followed by Dunnett’s test, N = 3.

**Table 1 antioxidants-12-00499-t001:** Organosulfur compound compositions in fresh Voghiera garlic hydro-alcoholic extract and after different storage periods (µg/g of fresh matter, n = 3 ± SD). * *p* < 0.05 vs. fresh garlic.

	Organosulfur Compounds	F (µg/g)	3 M +4 °C (µg/g)	3 M −4 °C (µg/g)	6 M −4 °C (µg/g)	3 M −4 °C + 3 M +4 °C (µg/g)	9 M −4 °C (µg/g)
1	Disulfide, bis (2-methylpropyl)	4.89 ± 0.10	2.63 ± 0.08 *	2.60 ± 0.04 *	1.65 ± 0.03 *	2.46 ± 0.04 *	2.94 ± 0.08 *
2	Diallyl sulfide	9.73 ± 0.19	3.75 ± 0.11 *	3.32 ± 0.05 *	2.38 ± 0.05 *	2.54 ± 0.04 *	1.07 ± 0.03 *
3	1,3-Dithiane	18.09 ± 0.36	54.56 ± 1.64 *	36.46 ± 0.55 *	8.02 ± 0.16 *	7.23 ± 0.12 *	6.39 ± 0.17 *
4	Methyl-1-propenyl disulfide	2.39 ± 0.05	19.33 ± 0.58 *	10.85 ± 0.16 *	1.41 ± 0.03	1.66 ± 0.03	1.17 ± 0.03
5	Disulfide, di-tert-dodecyl	111.27 ± 2.23	16.89 ± 0.51 *	13.08 ± 0.20 *	5.67 ± 0.11 *	6.80 ± 0.12 *	6.62 ± 0.17 *
6	Diallyl disulfide	190.03 ± 3.09	216.85 ± 6.51 *	196.68 ± 2.95	22.67 ± 0.45 *	10.01 ± 0.17 *	21.86 ± 0.57 *
7	Dipropenyl disulfide	10.01 ± 0.20	116.33 ± 3.49 *	89.10 ± 1.34 *	9.86 ± 0.20 *	0.28 ± 0.00 *	11.80 ± 0.31
8	Metoxymethyl isothiocyanate	30.97 ± 0.62	23.41 ± 0.70 *	16.86 ± 0.25 *	7.51 ± 0.15 *	7.66 ± 0.13 *	5.63 ± 0.15 *
9	Allyl methyl trisulfide	14.62 ± 0.29	33.80 ± 1.01 *	24.28 ± 0.35 *	3.76 ± 0.08 *	4.80 ± 0.08 *	4.18 ± 0.11 *
10	Isomer of Diallyl disulfide (1-propenyl allyl disulfide)	11.268 ± 1.25	1.57 ± 0.05 *	2.70 ± 0.04 *	3.73 ± 0.07 *	3.47 ± 0.06 *	0.56 ± 0.01 *
11	Isomer of Diallyl disulfide (1-propenyl allyl disulfide)	5.11 ± 0.10	4.22 ± 0.13 *	2.22 ± 0.03 *	2.39 ± 0.05 *	3.22 ± 0.05 *	0.94 ± 0.02 *
12	3-vinyl-1,2-dithiacyclohex-4 ene	455.78 ± 9.12	632.31 ± 18.97 *	539.51 ± 8.09 *	130.46 ± 2.61 *	139.71 ± 2.38 *	86.69 ± 2.25 *
13	di-2 propenyl trisulfide	40.59 ± 0.81	74.57 ± 2.24 *	52.89 ± 0.79 *	11.65 ± 0.23 *	15.62 ± 0.27 *	7.93 ± 0.21 *
14	3-vinyl-1,2-dithiacyclohex-5 ene	1342.27 ± 26.85	1890.63 ± 56.72 *	1566.05 ± 23.49 *	296.06 ± 5.92 *	317.52 ± 5.40 *	167.63 ± 4.36 *
	Sulfur compounds	2380 ± 47.60	3114.52 ± 93.44 *	2556.60 ± 38.35	507.22 ± 10.14 *	522.97 ± 8.89 *	325.42 ± 8.46 *

**Table 2 antioxidants-12-00499-t002:** Organosulfur compound composition in fresh Voghiera garlic diethyl ether extract (D-VGE) and after different storage periods (µg/g of fresh matter, n = 3 ± SD). * *p* < 0.05 vs. fresh garlic.

	Organosulfur Compounds	F (µg/g)	3 M +4 °C (µg/g)	3 M −4 °C (µg/g)	6 M −4 °C (µg/g)	3 M −4 °C + 3 M +4 °C (µg/g)	9 M −4 °C (µg/g)
1	Disulfide, bis (2-methylpropyl)	7.44 ± 0.17	4.69 ± 0.09 *	5.10 ± 0.11 *	2.22 ± 0.05 *	1.45 ± 0.03 *	3.42 ± 0.07 *
2	Diallyl sulfide	9.91 ± 0.23	6.23 ± 0.12 *	6.08 ± 0.13 *	2.32 ± 0.05 *	1.14 ± 0.02 *	2.69 ± 0.06 *
3	1,3-Dithiane	80.61 ± 1.85	16.56 ± 0.34 *	12.27 ± 0.26 *	40.38 ± 0.89 *	7.88 ± 0.16 *	38.45 ± 0.82 *
4	Methyl-1-propenyl disulfide	24.55 ± 0.56	2.88 ± 0.08 *	2.98 ± 0.06 *	18.48 ± 0.41 *	2.71 ± 0.05 *	14.40 ± 0.31 *
5	Disulfide, di-tert-dodecyl	358.03 ± 8.23	12.49 ± 0.21 *	13.43 ± 0.28 *	14.56 ± 0.32 *	5.44 ± 0.11 *	16.04 ± 0.34 *
6	Diallyl disulfide	290.03 ± 6.67	35.51 ± 0.69 *	37.67 ± 0.79 *	141.09 ± 3.10 *	47.56 ± 0.95 *	123.95 ± 2.65 *
7	Dipropenyl disulfide	105.60 ± 2.43	16.26 ± 0.32 *	17.90 ± 0.38 *	93.63 ± 2.06 *	28.21 ± 0.56 *	80.02 ± 1.71
8	Metoxymethyl isothiocyanate	56.37 ± 1.30	19.63 ± 0.38 *	13.31 ± 0.28 *	13.24 ± 0.29 *	7.86 ± 0.16 *	19.57 ± 0.42 *
9	Allyl methyl trisulfide	85.64 ± 1.97	17.56 ± 0.34 *	17.28 ± 0.36 *	33.89 ± 0.75 *	2.57 ± 0.05 *	22.91 ± 0.49 *
10	Isomer of Diallyl disulfide (1-propenyl allyl disulfide)	3.35 ± 0.08	4.32 ± 0.08 *	4.49 ± 0.09 *	36.32 ± 0.80 *	18.44 ± 0.37 *	1.49 ± 0.03 *
11	Isomer of Diallyl disulfide (1-propenyl allyl disulfide)	24.31 ± 0.56	1.54 ± 0.03 *	5.26 ± 0.11 *	2.60 ± 0.06 *	0.54 ± 0.01 *	2.12 ± 0.05 *
12	3-vinyl-1,2-dithiacyclohex-4 ene	1157.95 ± 26.36	465.26 ± 9.07 *	446.20 ± 9.37 *	430.46 ± 9.47 *	458.25 ± 9.16 *	427.19 ± 9.14 *
13	di-2 propenyl trisulfide	142.53 ± 3.28	59.14 ± 1.15 *	53.10 ± 1.12 *	67.67 ± 1.49 *	10.97 ± 0.22 *	42.88 ± 0.92 *
14	3-vinyl-1,2-dithiacyclohex-5 ene	3747.55 ± 86.19	1741.02 ± 33.95 *	1664.09 ± 34.95 *	1293.59 ± 28.46 *	1638.87 ± 32.78 *	1187.99 ± 25.42 *
	Sulfur compounds	6093.87 ± 140.16	3114.52 ± 93.44 *	2299.15 ± 48.28 *	2190.56 ± 48.19 *	2231.87 ± 44.64 *	1983.12 ± 42.44 *

## Data Availability

Not applicable.

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
