# Peer review of "The Effect of Different Storage Conditions on Phytochemical Composition, Shelf-Life, and Bioactive Compounds of Voghiera Garlic PDO"

_antioxidants, 2023, doi:10.3390/antiox12020499_

Round 1

Reviewer 1 Report

In the manuscript the authors examine the effect of storage on the phytochemical composition, biological effects and shelf-life of Voghiera garlic PDO. The authors presented this theme based on  43 cited articles, what seem to be adequate to the specific field of the manuscript.

The presented data are interesting, however I have some  comments.

Materials and Methods

While the selection of the MCF7 line has been well documented, which was dictated by the choice of the RAW 264.7 cell line in experiment?

In most of the experiments performed, the incubation time with VGE was 24 hours, and in the case of cell cycle distribution 72 hours. Please explain these discrepancies.

What was the protein content in the cell lysates for immunochemical analysis?

Results and discussion

I suggest adding representative immunoblots for the analysis of the levels of E-cadherin

Author Response

In the manuscript the authors examine the effect of storage on the phytochemical composition, biological effects and shelf-life of Voghiera garlic PDO. The authors presented this theme based on  43 cited articles, what seem to be adequate to the specific field of the manuscript.

The presented data are interesting, however I have some comments.

Materials and Methods

  • While the selection of the MCF7 line has been well documented, which was dictated by the choice of the RAW 264.7 cell line in experiment?

R. According to the Reviewer’s indication, we have added the following sentence in the manuscript: “RAW 264.7 cells are a macrophage-like, Abelson leukemia virus-transformed cell line obtained from BALB/c mice. This study has made use of this cell line as a typical and established model of mouse macrophages for the investigation of cellular reactions to microorganisms and their metabolites.” This sentence has been added at paragraph 3.4 Antioxidant Properties of VGE on RAW 264.7 Macrophage Cells.

  • In most of the experiments performed, the incubation time with VGE was 24 hours, and in the case of cell cycle distribution 72 hours. Please explain these discrepancies.

R. We thank the reviewer for her/his question. For evaluation of the effects of VGE on cell cycle distribution and E-cadherin expression in MCF7 cells, we chose a 72-hour treatment based on our previous evidence that this corresponds to the maximal effect of our fresh garlic extract in this breast cancer cell line (see ref.7, Brugnoli et al., 2021)

  • What was the protein content in the cell lysates for immunochemical analysis?

R. We thank the reviewer for her/his question, the protein content in the cell lysates was 15 µg, and we added it in the paragraph 2.11.

Results and discussion

I suggest adding representative immunoblots for the analysis of the levels of E-cahderin

 R. We thank the reviewer for her/his suggestion, we added a representative immunoblot of E-cadherin levels in Figure 3 (new Figure 3B).

Reviewer 2 Report

1. The manuscript is overloaded with descriptions of the importance of the Voghiera garlic PDO in the introduction. A strong and concise description of this section is needed.

2. Although the introduction falls within the aim and scope of the results the novelty needs substantial revision

3. 2.14- please provide the reference.

4. Line 33 - "were stable during the storage period" - please provide details .

5. The significance markers in Figure 2 do not clearly show the significant differences between the groups, please correct them.

6. Please explain the dotted lines in the figure in the figure caption of Figure 3.

7. The authors need to provide the number of experimental parallels (i.e. n values) in the manuscript on page 16, line 525-527 and 530-533.

8. Please standardize the nomenclature of RAW 264.7 cells throughout the manuscript on page 16, line 526 and 531.

9. The authors have used in vitro models to estimate the biological activity of Voghiera garlic PDO extract, but the tests appear to be sparse. Please clearly explain the linkage and rationale for conducting these assays.

10. Please describe and discuss the unique features of this research in the context of future applications and their influence.

11. All the references should be in one library. Please refer to Author's guide latest requirements for manuscripts. 

Author Response

The manuscript is overloaded with descriptions of the importance of the Voghiera garlic PDO in the introduction. A strong and concise description of this section is needed.

R. We thank the reviewer for her/his comment, accordingly we have reduced the description of the Voghiera garlic PDO.

Although the introduction falls within the aim and scope of the results the novelty needs substantial revision

R. We thank for this indication and, accordingly, in the revision version of the manuscript we better specified the contribution of our work

2.14- please provide the reference.

R. We thank the reviewer for the indication, we provided the reference

Line 33 - "were stable during the storage period" - please provide details .

R. We thank the reviewer for her/his comments, and we specified “the amount” of antioxidant compounds were stable during the storage period, as showed in figure 2 where flavonoids, total phenolic compounds and related antioxidant activity were not significantly decreased during the months of storage.

The significance markers in Figure 2 do not clearly show the significant differences between the groups, please correct them.

R. We thank the reviewer and we have corrected the position of the markers.

Please explain the dotted lines in the figure in the figure caption of Figure 3.

R. We thank the reviewer for the question. The dotted lines indicate the values of the control conditions (Vehicle), we amended the description of Figure 3A in the revised version of the manuscript.

The authors need to provide the number of experimental parallels (i.e. n values) in the manuscript on page 16, line 525-527 and 530-533.

R. Done, as requested.

Please standardize the nomenclature of RAW 264.7 cells throughout the manuscript on page 16, line 526 and 531.

R. Done, as requested.

The authors have used in vitro models to estimate the biological activity of Voghiera garlic PDO extract, but the tests appear to be sparse. Please clearly explain the linkage and rationale for conducting these assays.

R. As the garlic extract, due to its constituents, shows antitumoral, antioxidant and antiinflammatory effects, experiments have been designed to address all these aspects. Therefore, antitumoral activity was assessed in the breast cancer-derived MCF7 cell lines, while antioxidant and antiinflammatory activities were evaluated considering the capability of garlic extracts to reduce reactive oxygen species and nitric oxide considered an important regulator of inflammation. In fact, NO is a key signaling molecule involved in a variety of biological processes, including vascular and immunological function. In order to trigger a protective response, it specifically activates immune cells, particularly macrophages. However, its excessive secretion causes damage to central and peripheral illnesses, indicating that its control is required to maintain human health.

We have better specified these concepts in the Introduction section.

Please describe and discuss the unique features of this research in the context of future applications and their influence.

R. We thank the reviewer for her/his question. In the present work, an extensive investigation was conducted on the quali-quantitative changes of the chemical metabolites and related biological activities of Voghiera garlic PDO at different storage times.

Numerous studies have demonstrated that various factors pre- and post-harvest can affect the chemical composition of garlic, specifically organosulfur compounds, which are involved in pungency of garlic and in its bioactive effects.

To our knowledge, this is the first time that a multi-methodological approach has been applied to evaluate phytonutrients (not only organosulfur compounds but also the antioxidant fraction, flavonoids, polyphenols) at different time/temperature storage periods and try to correlate the chemical changes with two important effects of garlic such as reduce the breast tumor progression and the pro-inflammatory effect. Furthermore, the evaluation of phenolic compounds and its related antioxidant activity during time highlighted in the paper turn out to be important for human health and also for shelf life of the product and its uses in food industry.

This knowledge is important for the production of functional products which must be weal standardized in term of concentration of phytochemicals, for the use of garlic to reduce the oxidative process, given its high amount of antioxidant compounds also for long storage period; and for Voghiera garlic producers to evaluate the best method of preserving this vegetable, guaranteeing as much as possible all the important chemical compounds. We have better specified in the Conclusion section.

 All the references should be in one library. Please refer to Author's guide latest requirements for manuscripts. 

R. We thank the reviewer for her/his indication. We re write the references as indicated by the reviewer, using the Zotero software package as suggested in the author’s guideline.

Reviewer 3 Report

The subject matter of the original paper - Effect of different storage conditions on phytochemical composition, shelf-life and bioactive compounds of Voghiera garlic PDO. [2163599] has the potential for publication in Antioxidants in present form.

This paper contains interesting and useful results for garlic storage. Garlic has been shown to have significant and interesting antioxidant activity over long storage periods at a refrigerated temperature. Organosulfur compounds, contained in large quantities in fresh garlic, decreased in long storage times, with an important loss after 6 months.

The contribution is well structured, the materials and methods are quite informative to allow replication of the experiment, and the statistical methods used are correct and adequate. The results are clearly presented, the tables and figures are all necessary, complete and clearly presented. The references are adequate.

Author Response

The subject matter of the original paper - Effect of different storage conditions on phytochemical composition, shelf-life and bioactive compounds of Voghiera garlic PDO. [2163599] has the potential for publication in Antioxidants in present form.

This paper contains interesting and useful results for garlic storage. Garlic has been shown to have significant and interesting antioxidant activity over long storage periods at a refrigerated temperature. Organosulfur compounds, contained in large quantities in fresh garlic, decreased in long storage times, with an important loss after 6 months.

We thank the reviewer for her/his positive comments.

Round 2

Reviewer 2 Report

The article has been carefully revised as requested by the reviewer and may be published in the Antioxidants